# Bilateral Congenital Knee Dislocation in Colombia: Case Report and Literature Review

**DOI:** 10.3390/children10010020

**Published:** 2022-12-22

**Authors:** Jefferson Augusto Salguero-Sánchez, Santiago Andrés Sánchez-Duque, Ivan David Lozada-Martínez, Yamil Liscano, Jhony Alejandro Díaz-Vallejo

**Affiliations:** 1Departamento Materno Infantile, Facultad de Ciencias de la Salud, Universidad de Caldas, Manizales 170004, Colombia; 2Medical and Surgical Research Center, Future Surgeons Chapter, Colombian Surgery Association, Bogotá 110111, Colombia; 3Grupo de Investigación en Salud Integral (GISI), Departamento Facultad de Salud, Universidad Santiago de Cali, Cali 760035, Colombia

**Keywords:** knee, bilateral congenital knee dislocation, newborn

## Abstract

Congenital knee dislocation (CKD) is a rare disease with an estimated incidence of 1 per 100,000 live births, characterized by a rare musculoskeletal malformation in genu recurvatum deformity present at birth, affecting one or both lower limbs. The diagnosis may be suspected during ultrasound when observing that the situation of the extremities is not correct, and is confirmed by physical examination at birth, with plain radiography being helpful to establish the degree of severity. At present there are controversies regarding treatment and there is no definitive protocol. We present a new case of CKD, observed in the city of Manizales, diagnosed immediately after birth.

## 1. Introduction

Congenital knee dislocation (CKD) is a rare disease characterized by anterior and outward displacement of the tibia in relation to the femur. Three degrees of severity are recognized: congenital hyperextension of the knee, congenital hyperextension with anterior subluxation of the tibia over the femur and congenital hyperextension with anterior dislocation of the tibia over the femur. It was first described by Chanssier in 1812 and later by Chatelain in 1822 and Bord in 1834 [1,2]. In 1925, Kopits, reviewing patients with congenital anomalies, found that 65% of patients with CKD have other congenital anomalies such as hip dislocation and foot deformities [3]. McFarland in 1929 and O’Dell and Holt in 1954 found that in such patients the anterior cruciate ligament was elongated and underdeveloped or even absent [4,5]. In 1964, Finder classified congenital hyperextension deformities of the knee into five distinct clinical entities. The category to be considered in this entity is CKD [6]. In 1967, Katz proposed some possible causes of this pathology and confirmed the genetic transfer of this anomaly already proposed by Provenzano and McFarlaney in 1947 [4,7,8].

The known epidemiological information on this disease is not very extensive, but its incidence is estimated to be 1 per 100,000 live births [7], or approximately 1% of the incidence of developmental dysplasia of the hip [8]. It is twice as frequent in girls as in boys, and there is no difference between the right and left knee and one third of the cases are bilateral [9]. No predominance has been observed in any specific geographic region. In total, this pathology has been identified in more than 400 individuals [10].

The diagnosis is generally made at birth, given the position of the knee, which is confirmed radiologically [11]. For this reason, it is also important that the delivery personnel perform a complete physical examination to rule out other syndromes or associated anomalies [10]. According to the literature, this diagnosis has also been confirmed prenatally, two by radiography in 1986 [12] and 1993 [13], and another two by ultrasound in 2003 [14,15]; despite the cases mentioned, these prenatal descriptions are very rare [16].

A rare case of bilateral congenital knee dislocation (BCKD), diagnosed after birth, with a history of previous delivery with the same pathology, is presented. Therapeutic generalities and new treatment modalities under study are also presented.

## 2. Case Presentation

This study presents a female patient in her fourth decade of life, in her second pregnancy, with a medical history of congenital dislocation of the knee and a firstborn with the same pathology. Prenatal control ultrasounds were performed without alterations; there was adequate adherence to micronutrients; the pregnancy was uneventful and in week 40 a female baby was born spontaneously, with adequate weight and height, with a neonatal apgar score of 8 at one minute and 10 at five minutes, and adequate neonatal adaptation. The initial physical examination showed hyperextended knees associated with genu valgum and flat feet, corresponding to grade III; the rest of the examination was normal (Figure 1). The orthopedist later confirmed the diagnosis. Unfortunately, no genetic studies were ordered. Bilateral casts were placed on both knees after manual reduction and we continued orthopedic treatment with progressive reduction of the dislocated knee.

## 3. Discussion

As previously expressed, according to [17] the etiology may be due to extrinsic or intrinsic causes, which have been corrected and clarified over time. Within the intrinsic causes are genetic anomalies; pathologies with other neuromuscular disorders can cause this hyperextension, for example:Larsen’s syndrome. This describes patients with facial dysmorphism, joint hyperelasticity and multiple dislocations and whose frequency is 1 per 100,000 live births [18].Down’s syndrome. Patients may have muscular hypotonia and ligamentous hyperlaxity, which favors the appearance of conditions of this type [19].Arthrogryposis. This consists of congenital, non-progressive and symmetrical joint contractures affecting at least two different areas of the human body [20].Myelomeningocele, in the condition of paralysis [21].These have also been associated, but with less prevalence, with camptodactyly, Ehlers–Danlos syndrome, mongolism, cryptorchidism, angiomas, facial palsy and imperforate anus [1].

Among the extrinsic causes are mechanical anomalies, such as fetal malposition associated with oligohydramnios, primary contracture of the quadricipital tendon, anterior cruciate ligament malformation and dislocation during delivery [22]. This pathology may be associated with a polygenic hereditary component, due to the association along with other additional anomalies [1]. This was as in our case which had a familial component with the case of the first child; however, no associations with other pathologies have been found. This condition is characterized by contracture of the extensor mechanism of the quadriceps and of the anterior capsule of the knee joint, intraarticular adhesions, and hypoplasia or absence of the patella that begins to form after the dislocation is corrected. The supraratellar bursa is obliterated by tendon adhesions, the collateral ligaments are displaced forward, the hamstring muscles are subluxed forward (functioning as knee extensors), the iliotibial band is hyper rotated, the patella may be displaced outward, and the cruciate ligaments may be altered or missing altogether [10,23]. The congenital anomalies most frequently associated with congenital dislocation of the knee are developmental dislocation of the hip and congenital anomalies of the feet [1].

Clinical assessment should be performed by exploring the entire lower extremity to quantify the fixation and axis of both lower limbs, but diagnostic imaging is also necessary. Second trimester ultrasound would be indicated for prenatal diagnosis, where the relationship of the baby to the foot and head usually suggests a congenital dislocation. Radiography is useful to determine the degree of involvement, while computerized tomography (CT) and magnetic resonance imaging (MRI) are rarely used [9,15]. In our case, the diagnosis was made by physical examination, and direct double projection (postero-anterior and lateral) single radiography of the lower limbs was taken.

In 1946, Leveuf and Pais classified the deformity into three groups: hyperextension, subluxation and dislocation. In 2011, Abdelaziz and Samir made a classification based on range of motion, independent of joint relationship [9]. Currently, the most widely used classification is that of Laurence and Curtis Fisher, dividing CKD into three grades as follows [1,12,13]:Grade I. Represents hyperextension of the knee joint at birth without displacement of the articular surfaces of the femur in relation to the tibia (the axes of the long bones are opposite each other in the joint line).Grade II. Represents subluxation, the tibial epiphysis slipping on the anterior aspect of the femur over the articular surface of the condyle.Grade III. Represents total dislocation of the tibial epiphysis in front of the femoral condyles. In our case, the patient corresponded to a grade III.

Treatment should be performed immediately and is focused according to the level of involvement and degrees of passive flexion and coaptation of the joint. Manual reduction and flexion cast immobilization are recommended to be initiated within the first 24 h, especially for grade 1 and 2 deformities, and surgical treatment is recommended for grades 3, to loosen the quadriceps, iliotibial band, adductor and collaterals, and anterior cruciate ligament [24]. Treatment has also been proposed according to the degrees of bending as follows [24]:If >90° of passive flexion. Treatment will be by serial casts, which are maintained for about 2–4 weeks. Immobilization is performed under control with entry scopy without maintained traction.If flexion is 30° to 90°. Initially treated with weekly casts and mobility is re-evaluated after 4 weeks. If >90° is achieved, conservative treatment with casts is maintained, while a quadricipital tenotomy is recommended if flexion after those 4 weeks is still <90°.If flexion less than 90° is still maintained after tenotomy, V-Y plasty associated with arthrotomy is recommended. If more than 30° of passive flexion is not achieved, and in case of recurrences, a V-Y plasty associated with an arthrotomy would be performed to release the ligamentous structures that are displaced anteriorly.

## 4. Conclusions

BCKD is a rare disease of varied etiology, which can be diagnosed without difficulty and even prenatally if done accurately. However, an exhaustive search for other pathologies that may occur in these cases should be made, because the effectiveness of treatment is hindered by comorbid conditions, especially diseases with genetic syndromes that decrease the effectiveness of conservative measures. Treatment is usually effective if carried out early, but it depends on the degree of involvement and the condition of the structures involved in the deformity, and if it is not sufficient, the early initiation of more complex measures, in search of the least number of interventions possible, is still the best way to avoid complications and the appearance of comorbidities related to the condition. One must take into account the aesthetic impact for parents as a trigger that allows for accelerating the intervention process, and the need to follow-up between 3 and 5 months after the first intervention to prevent late identification of recurrence of the condition. It is recommended to generate a centralized registry of cases in Latin America in order to contribute to a discussion that favors early intervention and the identification of common characteristics in this patient population.

## Figures and Tables

**Figure 1 children-10-00020-f001:**
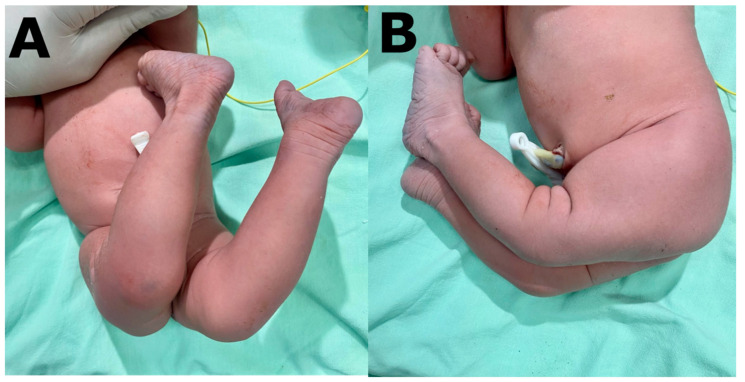
(**A**) Frontal view of a patient with BCKD, (**B**) lateral view of a patient with BCKD.

## Data Availability

Not applicable.

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
