# Peer review of "Bilateral Congenital Knee Dislocation in Colombia: Case Report and Literature Review"

_children, 2022, doi:10.3390/children10010020_

Round 1

Reviewer 1 Report

1.      What is “CSF” (line 33) an abbreviation for?

2.     In line 100, should “2nd trimester” be “Second trimester”?

3.     You mentioned “a direct double projection single radiography of the lower limbs was taken” (line104-105) and the patient corresponded to grade III, please add the images of X-ray in “Case Presentation”.

4.     What is the range of the motion of the knees? Which treatment did you choose and why? What is the duration of your follow-up and what was the outcome?

Author Response

Dear, we appreciate the comments that help us to improve our work. Below are the suggested corrections. We remain attentive.

  1. What is the abbreviation for “CSF” (line 33)?

corrected

  1. On line 100, should “second quarter” be “second quarter”? corrected

  1. You mentioned "simple x-ray of the lower limbs in double direct projection" was mentioned (line 104-105) and the patient corresponded to grade III, please add the x-ray images in "Case Presentation".

An attempt was made to contact the parents, however, as they came from a rural area, communication could not be made. However, we will keep it in mind for the next opportunity to improve our work.

  1. What is the range of motion of the knees? What treatment did you choose and why? What is the duration of your follow-up and what was the result? Corrected

Reviewer 2 Report

The pathology is already well described.

It is no novelty.

Author Response

Dear, we appreciate the comments that help us to improve our work. However, we considered this case to be important for Colombia since there are few reports, and it is also the first case of a girl from a rural area, which is noteworthy.

Reviewer 3 Report

This is a case report on a newborn with a rare and congenital disorder. English spelling and grammar can be improved.

Title: I would suggest against using an abbreviation in the title. Please find a better name for the title.

The case presentation per se could benefit from more data, and is mandatory for this manuscript.

E.g. What was the way of delivering the newborn? What was the birth score?

Was there any treatment instilled after birth? What could be some treatment options for patients with these disorders?

Are there any data regarding the family history? Do the authors have any history that can be relevant here?

Authors mention diagnostic tool as ultrasonography or radiography. Can authors provide any imaging for this? This could greatly add value to scientific literature

Did the mother/family members of the newborn sign any informed consent allowing these medical information’s to be published globally? If not, please explain why.

Please begin the discussion section with the most important finding of your case report.

Reference style are fine.

Author Response

Dear, we appreciate the comments that help us to improve our work. Below are the suggested corrections.

Title: I would suggest not using an abbreviation in the title. Find a better name for the title. corrected

The presentation of the case per se could benefit from more data and is mandatory for this manuscript. corrected

E.g. What was the way to give birth to the newborn? What was the birth score?

Female patient in her 4th decade of life, in her second pregnancy with a medical history of congenital dislocation of the knee and her firstborn with the same pathology. Prenatal control ultrasounds were performed without alterations, adequate adherence to micronutrients, the pregnancy was uneventful, in week 40 a female baby was born spontaneously, adequate weight and height, with a neonatal apgar score of 8 at one minute and 10 at five minutes, with adequate neonatal adaptation. 

Was there any treatment instilled after birth? What might be some treatment options for patients with these disorders?

An attempt was made to contact the parents, however, as they came from a rural area. However, we expected to perform the surgery with a one-month interval between knees with postoperative management consisting of immobilization with a 90° flexion cast for two months, then a cast tube with the knee in 45° flexion, and then 60° flexion anterior cast splints on each knee for intermittent use. 

Is there data on family history? Do the authors have any stories that might be relevant here?

An attempt was made to contact the parents, however, as they came from a rural area, communication could not be made.

The authors mention ultrasonography or radiography as a diagnostic tool. Can the authors provide any images for this? This could add a lot of value to the scientific literature.

An attempt was made to contact the parents, however, as they came from a rural area, communication could not be made.

Did the newborn's mother/family members sign any informed consent allowing this medical information to be published globally? If not, please explain why.

Yes, informed consent is obtained

Round 2

Reviewer 1 Report

Thank you and looking forward to the outcome of  your treatment.

Reviewer 2 Report

Though not very novel as an entitiy, perhaps a reminder of this rare disease is appropriate. The case itself is well presented.

Reviewer 3 Report

The authors should include their response inside the manuscript. Otherwise, I accept the changes made.